# Stem Cells from a Female Rat Model of Type 2 Diabetes/Obesity and Stress Urinary Incontinence Are Damaged by In Vitro Exposure to its Dyslipidemic Serum, Predicting Inadequate Repair Capacity In Vivo

**DOI:** 10.3390/ijms20164044

**Published:** 2019-08-19

**Authors:** Istvan Kovanecz, Robert Gelfand, Guiting Lin, Sheila Sharifzad, Alec Ohanian, Randy Ricks, Tom Lue, Nestor F. Gonzalez-Cadavid

**Affiliations:** 1Division of Urology, Department of Surgery, Harbor-UCLA Medical Center and Los Angeles Biomedical Research Institute, Torrance, CA 90502, USA; 2Department of Urology, David Geffen School of Medicine at UCLA, Los Angeles, CA 90095-1768, USA; 3Department of Medicine, Charles Drew University of Medicine and Science, Los Angeles, CA 90059, USA; 4Department of Urology, UCSF School of Medicine, San Francisco, CA 94143, USA

**Keywords:** dyslipidemia, muscle-derived stem cells, fat infiltration, apoptosis, wound closure, microRNA, myostatin, interleukin-6

## Abstract

Female stress urinary incontinence (FSUI) is prevalent in women with type 2 diabetes/obesity (T2D/O), and treatment is not optimal. Autograph stem cell therapy surprisingly has poor efficacy. In the male rat model of T2D/O, it was demonstrated that epigenetic changes, triggered by long-term exposure to the dyslipidemic milieu, led to abnormal global transcriptional signatures (GTS) of genes and microRNAs (miR), and impaired the repair capacity of muscle-derived stem cells (MDSC). This was mimicked in vitro by treatment of MDSC with dyslipidemic serum or lipid factors. The current study aimed to predict whether these changes also occur in stem cells from female 12 weeks old T2D/O rats, a model of FSUI. MDSCs from T2D/O (ZF4-SC) and normal female rats (ZL4-SC) were treated in vitro with either dyslipidemic serum (ZFS) from late T2D/O 24 weeks old female Zucker fatty (ZF) rats, or normal serum (ZLS) from 24 weeks old female Zucker lean (ZL) rats, for 4 days and subjected to assays for fat deposition, apoptosis, scratch closing, myostatin, interleukin-6, and miR-GTS. The dyslipidemic ZFS affected both female stem cells more severely than in the male MDSC, with some gender-specific differences in miR-GTS. The changes in miR-GTS and myostatin/interleukin-6 balance may predict in vivo noxious effects of the T2D/O milieu that might impair autograft stem cell (SC) therapy for FSUI, but this requires future studies.

## 1. Introduction

Female stress urinary incontinence (FSUI) is in women the most prevalent form of incontinence, well above overactive bladder or overflow incontinence, affecting over 200 million women worldwide [1,2]. It occurs mainly in postmenopausal women, during pregnancy, childbirth, and low estrogen level, severely impairing their quality of life [3]. Despite the earlier emphasis on the role of fascia weakness or pelvic muscle sarcopenia/dysfunction in FSUI [4,5], now it is widely accepted that a bladder urethral sphincter dystrophy/dysfunction may be the primary cause. It is also well established that SUI prevalence is predominant in patients with type 2 diabetes (T2D) and especially with obesity (T2D/O) [6,7]. Physio-therapeutic/medical treatments are not optimal, and surgical treatments are limited to certain patients [8].

Therefore, stem cell therapy that is particularly targeting the urethral sphincter has emerged for repairing both the striated muscle in the external sphincter and surrounding the internal sphincter, and the smooth muscle inside the internal sphincter [9]. Early failures were mainly due to inadequate implanted cells and even to questionable clinical trials that were retracted [10]. Recent clinical trials are ongoing, particularly using muscle-derived stem cells (MDSC), adipose-derived stem cells, and mesenchymal stem cells [11,12,13].

The translational basic science underlying these trials has been hampered by not very effective animal models of FSUI associated with T2D/O [14], although results are promising for MDSC therapy [15,16,17]. However, they utilize stem cells from non-T2D/O animals, while in human therapy, mostly autografts are used, of stem cells exposed long-term to the same milieu that damaged the target sphincter and that presumably would impair their tissue repair capacity. Our group has shown this with MDSC from long-term T2D/O animals (LD-MDSC), the aged Zucker fatty (ZF) male rats, exhibiting erectile dysfunction and severe dyslipidemia. When LD-MDSC were implanted into the penile corpora cavernosa, they failed to repair the erectile dysfunction and the underlying corporal fibrosis, in contrast to the MDSC from young animals with mild early T2D/O (ED-MDSC) that were effective [18].

The male LD-MDSCs were imprinted in vivo by a noxious global gene transcriptional signature (gene-GTS), accompanied by alterations in the GTS of microRNAs (miR-GTS) [19]. These changes were reproduced in vitro on ED-MDSC by short-term incubation with up to 5% of severely dyslipidemic serum, or lipid factors, but not by high glucose, and were associated with fat infiltration, apoptosis, and over-expression of the striated muscle mass inhibitor/pro-lipofibrotic agent, myostatin [18], also present in the corporal smooth muscle [20]. This suggested the use of in vitro incubations of MDSC to detect changes by dyslipidemia and predict whether MDSC may be impaired in vivo by long-term exposure to the T2D/O milieu, and thus possibly affect autograft stem cell therapy in the donor/receptor patient with T2D/O. 

In the current work, we aimed to study whether the in vitro induced stem cell damage by the dyslipidemic serum is gender-independent, i.e., it occurs in MDSC from a female rat model counterpart with T2D/O-FSUI [21], and may predict for further studies in women with FSUI whether (1) dyslipidemia impairs in vivo the endogenous in situ tissue repair of pelvic muscle or urethral sphincter dystrophies and the FSUI stem cell therapy with MDSC autografts implanted on those targets; (2) the miR-GTS may serve as biomarkers of stem cell damage and hence of their tissue repair capacity.

## 2. Results

### 2.1. miR-GTS showed Key Differences Between the Female and Male MDSC, and Between the Female MDSC from T2D/O and Normal Rats, Suggesting that miR-GTS might be MDSC Identity Biomarkers

Twelve weeks old female Zucker Fatty (ZF) rats (ZUC-FA/FA; Crl:ZUC-Lepr^fa^ 185), their Zucker Lean (ZL) counterparts (ZUC-LEAN; Crl:ZUC-Lepr^fa^186) of the same strain used for the study on male rats [18,19] were used here for the isolation of the female stem cells. The young rats’ age was intended to isolate stem cells that were as healthy as possible so that they could be tested in culture for damage caused by exposure to dyslipidemic serum from older rats.

The ZF male rats are a model of metabolic syndrome, starting around 3–4 months of age to evolve into frank T2D/O with moderate hyperglycemia and morbid obesity, associated with erectile dysfunction, diabetic nephropathy, and arteriosclerosis [18,19,22,23,24]. In contrast, in the female rats, the hyperglycemia and obesity are not initially present, and later on, are much less marked than in the males. The glycemia in the female rats was 104 mg/dl without visible serum hyperlipidemia at 12 weeks of age, staying at 111 mg/dl (93 mg/dl in ZL), but with intense turbidity due to hyperlipidemia at 24 weeks of age. This female rat model has been associated with FSUI denoted by symptoms that were detected at 12 weeks of age [25] and also at 16 weeks of age [21].

The female MDSCs used in this work are different from the male ones we previously reported from the strains mentioned above [18,19] in the following aspects: 1) the female MDSC originated from the pelvic floor levator ani muscle, not from the gastrocnemius; 2) they were not isolated by the pre-plating Sca-1 selection we described for Sca1+/CD34+ MDSC [18,19], but with a multiple selections ending in Sca1-/CD34+/CD45-/CD11b-/CD31-cells [26]. We named these female MDSC (to differentiate them from the male MDSC) as 1) ZF4-SC: from the ZF rats at 12 weeks of age, and 2) ZL4-SC: from the age-matched ZL rats. Similarly, the female rats’ sera are: 1) ZFS: turbid serum from the ZF rats at 22 weeks of age, and 2) ZLS: clear serum isolated from the ZL age-matched rats

Our previous reports showed that miR-GTS in the male MDSC were altered by in vivo exposure to the T2D/O milieu, or in vitro to the dyslipidemic serum and lipid factors [18,19], and acted as stem cell damage biomarkers. We have now determined first whether miR-GTS show key differences between the female and male MDSC and then between the female MDSC from T2D/O and normal rats, thus acting as stem cell identity biomarkers.

To facilitate comparisons among male and female MDSC, the miR-GTS are presented in Table 1 as per thousand miR compositions in all MDSC without treatment (C) and compared them to each other to determine differences among themselves. The legend is at the table bottom.

Remarkably, just three miRs (miR-143-3p, miR-99b-5p, miR-10a-5p) made 60.5% of the total miRs, and them together with the next 12 miRs reached 87.3% of the total 47 selected miRs, i.e., were representative of the respective miR-GTS for ZF4-SC-C. When these top 15 were compared as ratios to the male ED-MDSC-C values, four in the ZF4-SC were > 2 (miR-143-3p; miR-125a-5p; miR-221-5p; miR-222-3p) and two were < 0.5 (miR-100-5p and miR-100-5p). There was a considerable difference between the non-dyslipidemia exposed female ZF4-SC and male ED-MDSC that might be due to gender, the striated muscle of origin, or the isolation procedure of the MDSC.

These differences remained on comparing ZL4-SC-C with ED-MDSC-C, including the highest expressed miR-143-3p (except for one miR), with three new added. In turn, a comparison of the two non-exposed female stem cells (ZF4-SC-C/ZL4-SC-C ratios) showed some key differences, particularly predominantly higher ratios in the ZL4-SC. The ZF4-SC appeared different from the ZL4-SC, as expected, because one was from the genetically diabetic ZF strain (the Lepr^fa^ 185), and the other was from its related non-diabetic ZL strain (the Lepr^fa(+/-?)^186). Altogether, this suggested that miR-GTS might be identity biomarkers in the non-exposed MDSC.

### 2.2. The Female ZF4-SC miR-GTS for Myostatin-Related miRs and some Unrelated miRs were Affected by in vitro Exposure to Dyslipidemic Serum Similarly to the Reported Male ED-MDSC miRs, but the ZL4-SC Were Less Sensitive.

To assess whether the in vitro response of the female ZF4-SC and ZL4-SC to dyslipidemia was equivalent to the one in vitro (and also in vivo to T2D/O) of the male ED-MDSC, we investigated here the female MDSC reaction to the highly turbid dyslipidemic ZFS from the 24 weeks old ZF female rats. They had mild hyperglycemia and were already obese (578 g bodyweight vs. 263 g in the ZL rats). The ZLS from the age-matched non-T2D/O female ZL rats was used as a reference.

We had reported that the lipofibrotic and muscle mass inhibitor, myostatin, was overexpressed in male MDSC undergoing in vivo damage by T2D/O [19] and in vitro by dyslipidemic serum [18], in concordance with down-regulation of miRs that counteract myostatin expression in vivo. Therefore, we tabulated, as a reference in Table 2 (second column), 12 of the down-regulated myostatin-related miRs, previously reported for the male MDSC in vivo, as LD-MDSC/ED-MDSC ratios (LD/ED) (legend at the table bottom) This was done to compare the in vivo effects of the T2D/O milieu on the male MDSC with the in vitro changes induced by the dyslipidemic serum on the female MDSC (remaining five columns). The in vivo male LD/ED values were organized in decreasing order of the in vitro expression values in the ZF4-SC-C. The same order was maintained with all ratios for the female MDSC treated by the rat sera versus their control without addition.

Remarkably, seven miRs, including the two most expressed ones (miR-21-5p and miR-199-5p), were down-regulated by 5% ZFS addition, but not changed by similar ZLS addition. Likewise, none of these 12 miRs was changed in the ZL4-SC by ZLS addition, or even by ZFS addition. Interestingly, both miR-21-5p and miR 199-5p values per thousand presented in Table 1 were approximately similar in the untreated ZF4-SC and ZL4-SC controls but responded differently to the treatment.

A similar tabulation of the remainder non-myostatin related miRs that had been changed >2, up or down in the reference male LD/ED ratios, is presented in Table 3 (legend at the table bottom), where eight out the 20 miRs down-regulated in the male stem cells by T2/O were also down-regulated in vitro in the female ZF4-SC by ZFS (none by ZLS), like in Table 2. Two of the three highest expressed miRs in the untreated ZF4-SC-C in Table 1, 99b-5p, and 10b-5p were severely down-regulated by ZFS, suggesting a key role in the ZF4-SC response to the dyslipidemic serum. The ZF4-SC were again more resistant to ZFS, since only two out of 20 tabulated miRs were down-regulated, and even two were up-regulated. In summary, the in vivo T2D/O effects on the miR-GTS of the male ED-MDSC were also reproduced in vitro in the female ZF4-SC (from Lepr^fa^ 185) but to a much lesser extent into the non-diabetic ZL4-SC lacking the leptin mutation.

### 2.3. miR-GTS Changes in the ZF4-SC by ZFS were Accompanied by Intracellular Fat Deposits and Apoptosis that in the ZL4-SC Occurred Similarly to ZF4-SC for Fat Deposits, and less for Apoptosis

Resembling what we reported on the male ED-MDSC [18], the in vitro incubations of the female ZF4-SC with the dyslipidemic ZFS at 5% caused a nearly 80-fold increase in the Oil red O staining for fat deposits versus the control or the ZLS addition (as shown in Figure 1 (please notice logarithmic scale).

Irrespective of the difference on some of the effects of ZFS on the ZL4-SC vs. ZF4-SC miR-GTS, Figure 2 (logarithmic scale) shows the same degree of fat infiltration occurring in both stem cells.

In turn, the apoptotic index was nearly 4-fold increased by ZFS acting on the ZF4-SC as compared to ZLS that was essentially similar to no addition (Figure 3, linear scale).

The ZFS effects were lower on ZL4-SC, about 3-fold, linear scale when compared to C, and only 1.5-fold when compared to ZLS (Figure 4 linear scale), suggesting less specific susceptibility for ZL4-SC than for the ZF4-SC to the effects of ZFS. The fact that ZLS increased apoptosis by 60% might indicate some general susceptibility to the serum of the ZL4-SC, that was significantly increased by exposure to ZFS.

### 2.4. The Female Stem Cell Damage Exerted in vitro by ZFS was also Accompanied by Inhibition of in Vitro Scratch Healing Repair in both the Male and Female MDSC, with the ZF4-SC being the Most Affected and the Male MDSC the Most Resistant

The determination of the effects of the hyperlipidemic serum on in vitro scratch healing by MDSC provides an in vitro approximation onto how this process may affect their migration and extracellular matrix formation in wound healing in vivo [27,28,29]. Figure 5 shows the not previously reported inhibition by ZFS of in vitro healing using male ED-MDSC as a reference, to compare now with the female counterpart MDSC. Panel C shows the initial scratch in ED-MDSC-C culture with added ZFS, with a gap separating both edges at the experiment initiation, and panel D, the closure exerted by their migration at 24 h. Panel A represents the quantitative assessment of the gap width, and Panel B of the % gap closure over a 48 h period. There was only 35% final gap closure (B) under ZFS addition, that was very specific since the ZLS addition allowed 100% gap closure, the same as no addition.

As to scratch healing by the female MDSC, two facts emerged, as shown in Figure 6. First, the untreated ZF4-SC and ZL4-SC were rather inefficient in comparison to the untreated male ED-MDSC in healing the scratch even after 72 h, as shown by the 55% gap closure for ZF4-SC and 60% for ZL4-SC. 

Second, at this time the damage by ZFS inhibited ZF4-SC completely (3% gap closure), and even the ZLS interfered partially (40% closure), whereas ZFS inhibited less the ZL4-SC (20% gap closure), with ZLS reaching even better closure than the no addition. So, both female MDSC were severely inhibited by ZFS, but the ones from the non-T2D/O animals (ZL4-SC) were more resistant to damage than the ZF4-SC from the early diabetic animals.

### 2.5. ZFS Induced in vitro Myostatin Over-Expression in both the ZF4-SC and ZL4-SC, as Previously Reported for the Male ED-MDSC, and this was Accompanied by the Inhibition of Interleukin-6, a Myostatin Counteractive Agent

To confirm whether myostatin was over-expressed by the dyslipidemic ZFS, as previously found in vitro in the ED-MDSC, replicating the in vivo pattern in LD-DSC [18,19], ZL4-SC and ZF4-SC were incubated not just with 5%, but 2.5% and 1% ZFS or ZLS, or in their absence (C). After 4 days, MDSC were collected, washed, and subjected to western blotting with antibodies for myostatin and beta-actin as a housekeeping protein. Figure 7 shows that in the various serum concentrations, myostatin was over-expressed by ZFS in the ZF4-SC versus C, and in the ZL4-SC, albeit to a lesser extent under 2.5% and 1% ZFS. The 1% ZLS concentration was unexpectedly as effective as ZFS, suggesting a basal, unspecific effect of rat serum unrelated to dyslipidemia. In addition to the similarity of ZF4-SC and ED-MDSC responses, this was concordant with ZFS down-regulation of miRs known to counteract myostatin in Table 2.

In turn, myostatin is known to be counteracted by interleukin-6 (IL-6) [30,31,32], and Il-6 was not expressed in the absence of added serum, but over-expressed by 2.5% of non-diabetic ZLS. This effect was fully inhibited by 2.5% ZFS, both on ZF4-SC and ZL4-SC, suggesting that ZFS blocked IL-6 by inducing myostatin over-expression.

## 3. Discussion

Our previous reports [18,19] had shown that the in vitro damage of male ED-MDSC by their short time exposure to 5% highly dyslipidemic serum: 1) mimicked their in vivo damage suffered by the LD-MDSC, long-term exposure to a highly dyslipidemic T2D/O milieu, and that 2) the in vitro miR-GTS alterations and other noxious effects might predict the impairment of their tissue repair capacity to the in vivo dyslipidemia effects. In this current paper, we have established the proof of concept that the in vitro damage process occurred with essentially similar features in a related MDSC (ZF4-SC) from young female counterpart rats with mild early T2D/O, and to a lesser extent in other MDSC (ZL4-SC) from rats without T2D/O. This implied that this in vitro process by ZFS on the female ZF4-SC might predict their in vivo damage by long-term T2D/O, similarly to the male ED-MDSC/LD-MDSC in vitro/in vivo correlation [18,19]. The in vitro susceptibility of MDSC to dyslipidemia seemed to be independent of gender since it occurred both in male and female MDSC. Moreover, it seemed to be unrelated to the diabetic Lepr^fa^ mutation since the MDSC damage was found also in the ZL4-SC that lack it, although the latter’s sensitivity to damage was lower. This assumption still requires experimental validation.

Our current main findings for the miR-GTS were first their potential validity for identifying and differentiating still normal or untreated/unexposed MDSC, an assumption supported by the key miR differences between the female ZF4-SC and ZL4-SC in comparison to the male ED-MDSC; some varying >2 fold (e.g., lower 100-5p, 99a-5p; higher 143-3p, 221-5p), as well as between the MDSC from the female diabetic and normal strains.

Second, the confirmation and expansion of their putative biomarker usefulness to follow up the T2D/O-induced damage on the exposed MDSC; in this case, the miR-GTS for the female ZF4-SC with ZFS. This applied particularly to the myostatin-related miRs (e.g., 21-5p, 199a-5p) [33,34] or unrelated (e.g., 99b-5p, 10b-5p) [35,36], which were affected similarly by exposure to their dyslipidemic ZFS as the male ED-MDSCs were with its dyslipidemic serum, and the finding that ZF4-SC were more sensitive than ZL4-SC. The few discrepancies remaining for the female ZF4-SC exposed in vitro to ZFS versus the male MDSC exposed in vivo to the T2D/O milieu could be a feature of these specific miRs or another manifestation of gender differences.

If these miR-GTS features are confirmed with stem cells other than MDSC in the rat, and then extended to humans, it may provide a specific biomarker to identify/categorize stem cells. Even gender differences may still be detected by miR-GTS since our comparison here may be affected by the isolation procedure and muscle of origin differences. Moreover, the use of miRs to clarify the role of myostatin in the stem cell damage may be based on what is known on its miR inter-regulation and the myostatin interaction with IL-6, as discussed below.

The in vitro induction of fat deposition and apoptosis in the female ZF4-SC by ZFS, and not by ZLS, resembling what happened in the male MDSC, and the ZFS inhibition of their migration and scratch closure (much higher than in male MDSC), denoted that both types of MDSC with a diabetes/obesity-prone gene mutation were susceptible to dyslipidemic milieu damage. Gender impact seemed to be significant for scratch closure since the female stem cells were more impaired than the male stem cells, and thus, this might translate into less effective wound-healing repair [27,28,29]. This requires further research to relate it with dystrophic pelvic muscle or urethral sphincter stem cell repair [37,38].

The over-expression of myostatin in ZF4-SC by ZFS, but also a lower but significant one by ZLS (absent in ZL4-SC), and the role of dyslipidemia in this process needs to be clarified in future studies vis-à-vis quantitative determinations of lipid factors in both types of serum. This may allow understanding the possible additional contribution of non-lipid factors. In any case, the ZFS-induced over-expression agreed with what we had reported previously with the male MDSC [18,19], and might be a key to clarify the mechanism of MDSC damage and its relevance to fat deposition [39,40,41] and even to potential induction of abnormal differentiation [42]. Moreover, it might suggest a putative anti-myostatin approach [43,44] to reverse/impede the damage, thus favoring both spontaneous repair and stem cell therapy for FSUI.

Based on the well-known involvement of myostatin in adipogenesis [45,46], apoptosis [47], and interference with wound healing [48,49], we might assume that myostatin may have an important role in these three processes. However, the demonstration would require to prove a modulation of the ZFS noxious effects on ZF4-SC and ZL4-SC by myostatin blockade vs. added recombinant protein.

Our finding that ZLS overexpressed IL-6 in both stem cells, but ZFS did not, might indicate that myostatin over-expression by ZFS (not occurring with ZLS) might inhibit IL-6, in contrast to what has been postulated in muscle cells [31,32,49]. This requires confirmation because of the key significance of the myostatin/IL-6 balance in the fat infiltration/apoptosis/failure of injury repair occurring in obesity and diabetes where stem cell damage may be a key.

As in the MDSC from the diabetic and very obese male rat model, collectively these changes in the counterpart female MDSC might predict the impairment of their tissue repair capacity by their exposure to the milder but still noxious in vivo T2D/O milieu. This potentially might render these stem cells partially or totally ineffective if implanted back for FSUI stem cell therapy. Therefore, as a model for the human stem cell damage, it is important to define whether this damage does make stem cells, and specifically the MDSC, in vivo-compromised for repairing FSUI in the animal, as in TD2/O-associated limb ischemia in a mouse model [50]. Perhaps more important is whether in rat models, this stem cell damage leads to abnormal stem cell lineage commitment [51,52,53]. If so, the risks would exceed inefficacy by causing the appearance of unexpected differentiated cells instead of the ones intended to be renewed.

Our findings agree with the interpretation of discrepancies in efficacy and desired outcomes in human stem cell clinical trials, based on the assumption that the disease status of patients may compromise the use of stem cells, and the need to define the characteristics of stem cells for assuring the safety and efficacy of their use in therapy [54]. This appears to have special relevance, particularly, for autografts of stem cells in diabetes patients, where these cells are affected in their mobilization and other features by the diabetic milieu [55,56].

For an eventual translation of our findings to human stem cell therapy for FSUI, it is necessary to demonstrate first in vitro that the human dyslipidemic serum can induce the noxious changes on the human stem cell of interest, and second that this is reflected in their miR-GTS. The current paper may provide a feasible and fast approach to this translation

## 4. Materials and Methods

### 4.1. Stem Cells and Serum Isolation

The female ZF and ZL rats were housed and treated according to The National Institutes of Health guides and used for isolating serum and stem cells under IACUC approval. To isolate the stem cells [57], the muscle tissue was enzymatically dissociated, first with collagenase and then dispase, after which non-muscle tissue was gently removed under a microscope. The cell suspension was filtered through a Falcon nylon filter (ThermoFisher Scientific, Waltham, MS, USA) and incubated with the following biotinylated antibodies: CD45, CD11b, CD31, and Sca1 (BD Biosciences, San Jose, CA, USA). Streptavidin beads (Milteny Biotec, San Diego, CA, USA) were then added to the cells together with antibodies for integrin-α7–phycoerythrin and CD34–Alexa647 (eBioscience, San Diego, CA, USA), followed by magnetic depletion of biotin-positive cells. The CD45-/CD11-/CD31-/Sca1-/CD34+/integrin-α7+ population was then enriched twice by flow cytometry (Becton- Dickinson, San Diego, CA, USA).

ZF4-SC and ZL4-SC were cultured in 0.1% gelatin-coated culture flasks in DMEM (4.5 g/L glucose), 10% FBS, 1% non-essential amino acids, 1% Na-Pyruvate (GE Life Sciences, Marlborough, MA, USA), and 1% Anti-Anti, and used in the 10th-15th passage. Reagents were from Gibco Life Technologies (Waltham, MA, USA).

### 4.2. MDSC Incubations and Scratch Wound Assay

ZF4-SC and ZL4-SC were incubated [18] (initial 40% confluence) for 4 days on collagen-coated 6 or 12-well plates, or 8-removable compartment slides, without addition (control: C) or adding ZFS or ZLS to 1–5%. The medium was then discarded, MDSC were washed with PBS and subjected to fixation for histochemistry, or fresh protein isolation for western blots, or RNA isolation for gene/miR-GTS [18]. For the wound assay [27], briefly, MDSC treated as above were plated on 12-well plates to generate 100% confluence in 24 h. Then, a scratch was made through the cell monolayer by pressing a 200 µL pipette tip against the bottom of the well. The detached cells and medium were removed, fresh medium was restored, and a picture was taken under the inverted microscope. The cells were left incubating, and at 4, 24, 48, and 72 h pictures were taken to follow wound closure. The distance between both sides of the wound was measured, and the time to a complete closure was determined.

### 4.3. Quantitative Histochemistry

MDSC on the 12 well plates were stained with Oil Red O for fat droplets [18,19], and cells on the 8-well slides were stained for TUNEL assays for the apoptotic index [19,20,58]. Quantitative image analysis (QIA) [18,19] was performed by computerized densitometry (100–200× magnification), on as many fields as necessary to cover the wells, followed by QIA.

### 4.4. Western Blots 

Immuno-detection on the membranes [18,19] was with primary antibodies (Santa Cruz Biotechnology, Santa Cruz, CA) against myostatin (GDF8/11, mouse monoclonal antibody SC-393335); interleukin 6 (IL-6, rabbit polyclonal antibody SC-1265-R); and housekeeping beta-actin (mouse monoclonal, followed by secondary antibodies: anti-mouse IgG, horseradish peroxidase (HRP)-linked antibody (Cell Signaling Technology, Danvers, MA, USA), or anti-rabbit IgG linked to HRP (Amersham GE, Pittsburgh, PA, USA). Bands were visualized using luminol (SuperSignal West Pico; Chemiluminescent, Pierce, Rockford, IL, USA). For negative controls, the primary antibody was omitted. Densitometric analysis was performed in certain cases as stated, correcting by the housekeeping proteins.

### 4.5. Global miR-GTS 

RNA was isolated [18] from ZF4-SC and ZL4-SC with mirVana™ miRNA isolation kit (Ambion, ThermoFisher, San Diego, CA, USA), determining quality by the Agilent 2100 Bioanalyzer (Agilent Technologies (Dako) Carpinteria CA, USA). miR content was estimated by Norgen Biotek Corporation (Thorold, ON, Canada) by next-generation sequencing for all miRs listed in the Sanger miRBase Release 18.0. Values were expressed per 10 million reads. Control values with no serum addition (C) were tabulated for the ZF4-SC (Table 2 and Table 3), and then in vitro treatment ratios against C were calculated for samples receiving ZFS or ZLS. Only miR ratios up- or down-regulated by at least 2-fold were selected unless stated. 

To compare them in the female ZF4-SC and ZL4-SC with the in vivo male MDSC, the previous in vivo ED/LD MDSC ratios [19] were included as a reference. The ED-MDSC were from 12 weeks old male ZF rats, previously named OZ [19], with mild hyperglycemia/dyslipidemia, and moderate overweight; while the LD-MDSC were from aged 32 weeks old male ZF rats, with moderate hyperglycemia, high dyslipidemia, and morbid obesity. In Table 1, miR values were calculated as per thousand of the total miRs in the respective specimen without treatment, independent of the total raw reads. The miR-GTS complete results are in the GEO library, as GSE134340.

### 4.6. Statistical Analysis

When applicable, values were expressed as mean ± SEM. The normality distribution of the data was established using the Wilk–Shapiro test. Multiple comparisons were analyzed by single-factor ANOVA, followed by post hoc comparisons with the Tukey multiple comparison test.

## Figures and Tables

**Figure 1 ijms-20-04044-f001:**
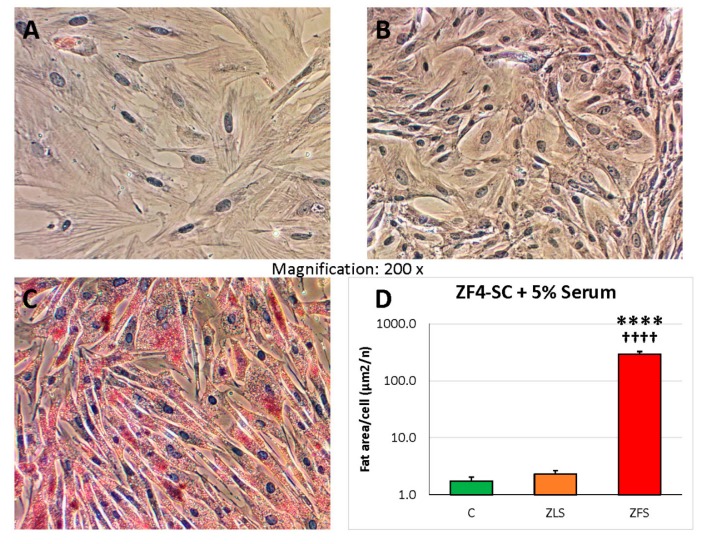
Incubation of ZF4-SC with ZF (Zucker fatty) serum (ZFS) induced intracellular infiltration by fat globules, whereas the one caused by ZL (Zucker lean) serum (ZLS) was very low and similar to the one in control with no addition. ZF4-SC were incubated for 4 days with no addition (**A**), or with added 5% ZLS from 20 weeks old ZL rats (**B**) or 5% ZFS from age-matched rats (**C**), and stained at 4–5 days with Oil Red O for fat infiltration. Pictures were taken at 200X, but QIA was applied to multiple fields at 100× (**D**) showing the bar graph of red area (fat) per cell, in a semi-logarithmic scale, C: no serum addition; **** *p* ≤ 0.0001 (C vs. ZFS); ^++++^
*p* ≤ 0.0001 (ZFS vs. ZLS).

**Figure 2 ijms-20-04044-f002:**
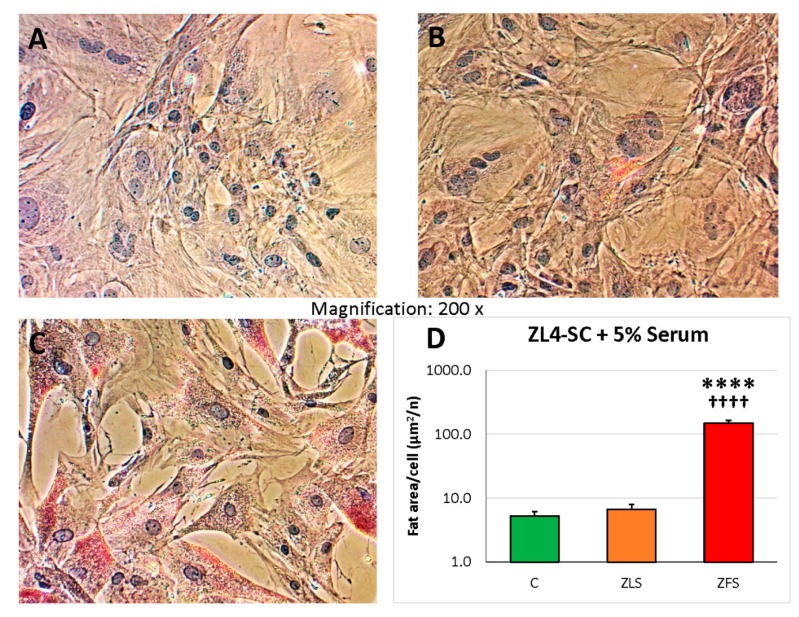
The fat infiltration of ZL4-SC caused by ZFS was similar to the one experienced by the ZF4-SC in Figure 1, with negligible infiltration by ZLS, like in no rat serum addition, indicating a similar response of both MDSC to ZFS. ZL4-SC were incubated as the ZF4-SC in Figure 1 with no addition (**A**), or added ZLS (**B**), or ZFS (**C**), then stained with Oil Red O, and pictures were taken as in Figure 1. (**D**) The bar graph is equivalent to the one in Figure 1D, with **** *p* ≤ 0.0001 (C vs. ZFS); ^++++^
*p* ≤ 0.0001 (ZFS vs. ZLS).

**Figure 3 ijms-20-04044-f003:**
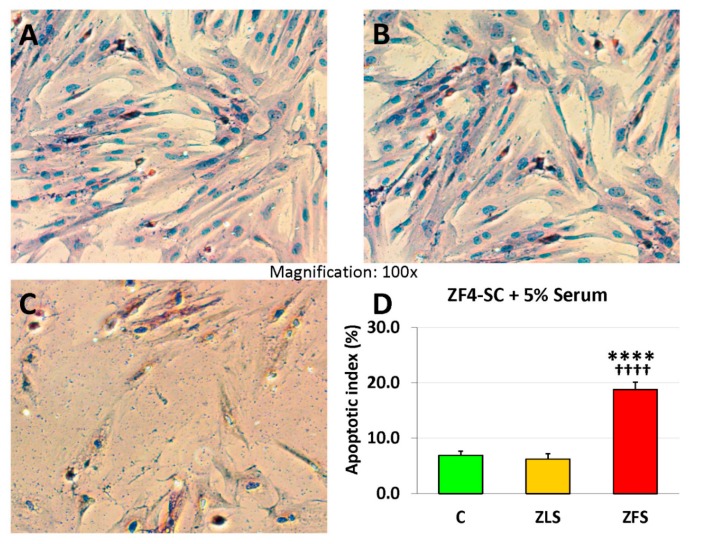
Incubation of ZF4-SC with ZFS-induced apoptosis, but the one caused by ZLS was negligible, similar to when no rat serum was added. ZF4-SC were incubated with no addition (**A**), or added ZLS (**B**), or ZFS (**C**), as in Figure 1, then subjected to the TUNEL reaction, and pictures were taken at 100X. The bar graph in (**D**) is equivalent to the one in Figure 1D, but on a linear scale with **** *p* ≤ 0.0001 (C vs. ZFS); ^++++^
*p* ≤ 0.0001 (ZFS vs. ZLS).

**Figure 4 ijms-20-04044-f004:**
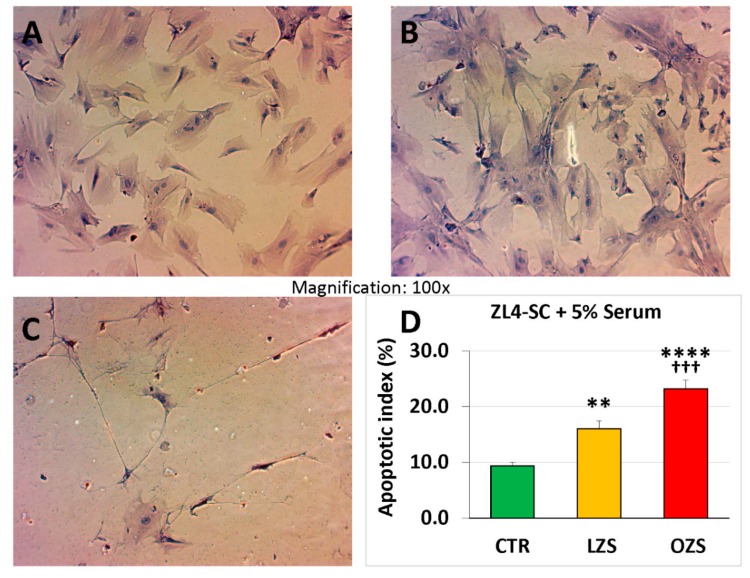
ZL4-SC were more sensitive to apoptosis caused by ZFS than the ZF4-SC shown in Figure 3, but they were also mildly affected by ZLS. ZL4-SC were incubated, as in Figure 1, with no addition (**A**) or added ZLS (**B**) or ZFS (**C**), and then subjected to TUNEL reaction, and pictures were taken at 100×. (**D**) The bar graph is equivalent to the one in Figure 1 but on a linear scale. C: no serum addition; **** *p* ≤ 0.0001 (C vs. ZFS); ** *p* ≤ 0.01 (C vs. ZLS); ^+++^
*p* ≤ 0.001 (ZFS vs. ZLS).

**Figure 5 ijms-20-04044-f005:**
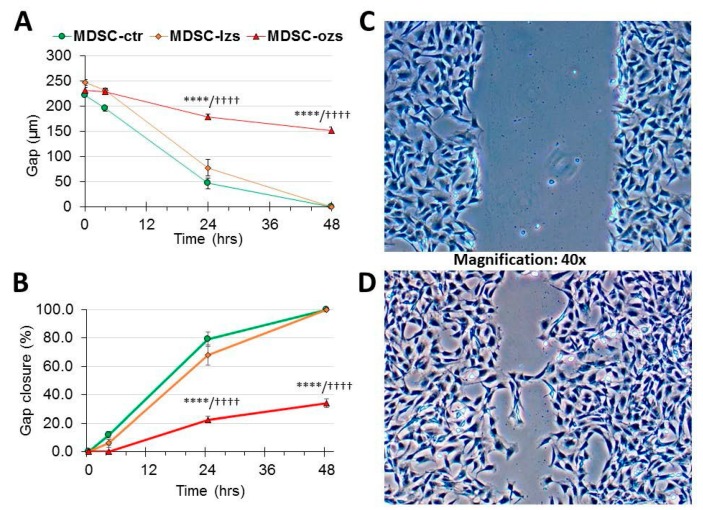
Gap closure of in vitro scratch injury by male ED-MDSC at 48 h was partially abrogated by the male lean Zucker serum, and it did not occur by the addition of the male obese Zucker serum, thus showing MDSC damage specifically by the latter. In this case, pertaining to the male ED-MDSC isolated in our studies of refs #18 and #19, and using their male lean Zucker serum (lzs), or obese Zucker serum (ozs), or no addition control (ctr), these preceding abbreviations are used, differing from the ones for the female rats used elsewhere (including Figure 6), to emphasize that both the stem cells and the sera were from male animals and not from their female counterparts. A gap was created in confluent monolayers of ED-MDSC, and either ozs or lzs was immediately added to 5% or not. Pictures were immediately taken at 0 h, and then at 3, 24, and 48 h, measuring both the gap width (µm) and its closure (%). Panel (**C**) is a micrograph at 0 h in the presence of added ZFS, and Panel (**D**) is the final one at 48 h with still a residual gap. Panel (**A**) is a time plot of the gap width, and Panel (**B**) is for gap closure. **** *p* ≤ 0.0001 (ctr vs. ozs), and ^++++^
*p* ≤ 0.0001 (lzs vs. ozs).

**Figure 6 ijms-20-04044-f006:**
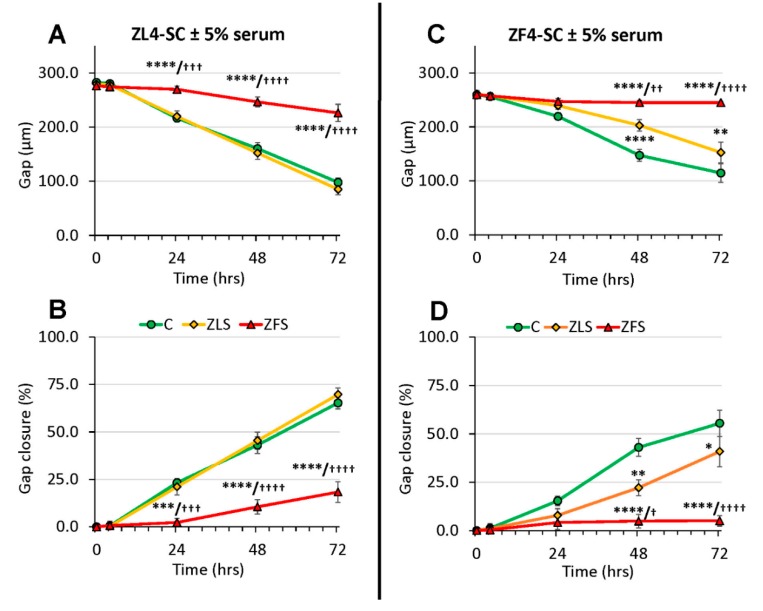
The efficacy of gap closure by ZF4-SC after gap injury was lower than the one for ED-MDSC, and it was more damaged by ZFS, but ZL4-SC were less affected than ZF4-SC. Experiments and measures were as in Figure 5, but with the female stem cells, and the female abbreviations. Panels (**A**) and (**C**): ZL4-SC; Panels (**B**) and (**D**): ZF4-SC; **** *p* ≤ 0.0001 (C vs. ZFS), *** *p* ≤ 0.005 (C vs. ZLS), ** *p* ≤ 0.01 (C vs. ZLS), and ^+^
*p* ≤ 0.05 (ZLS vs. ZFS), ^+++^
*p* ≤ 0.005 (ZLS vs. ZFS), and ^++++^
*p* ≤ 0.0001 (ZLS vs. ZFS).

**Figure 7 ijms-20-04044-f007:**
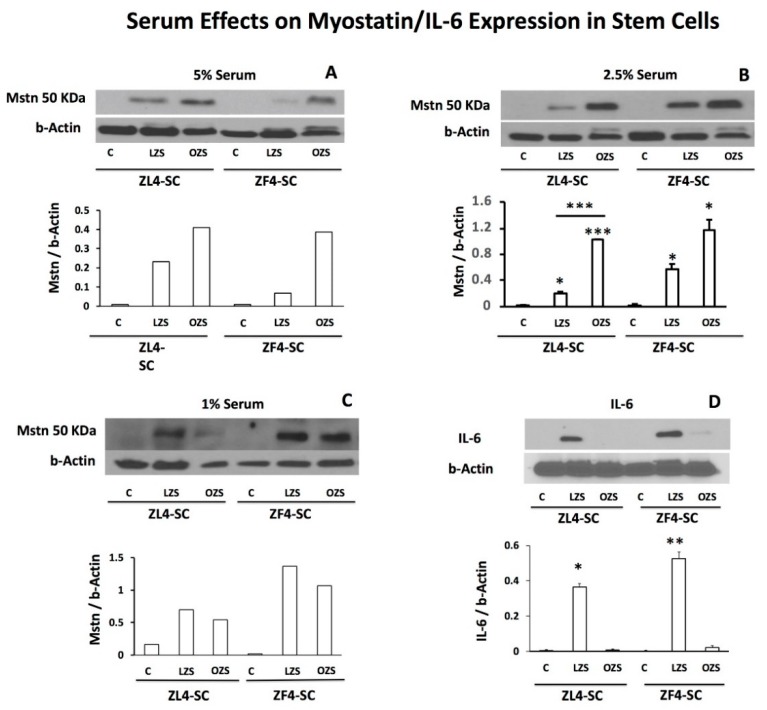
Myostatin in ZF4-SC and ZL4-SC was increased by ZFS (OZS) added to 5% and 2.5%, but little by ZLS (LZS), and the effects were less specific at 1%. This was associated with a potential counteraction by ZFS (OZS) of IL-6 expression. The respective stem cells were incubated in vitro for 4 days with either no addition, or with ZFS (OZS) or ZLS (LZS) added to 5% (**A**), 2.5% (**B**,**D**), or 1% (**C**), and the cell protein homogenates were assayed by western blot. The 50 kDa band was quantitated and corrected by the housekeeping gene beta-actin. Separate western blots were done for interleukin-6 (IL-6). Statistics were not applied on panels A and C, but based on this comparison, the effects of 2.5% incubations were repeated for statistical analysis as shown in panels B and D. *** *p* ≤ 0.001 (ZFS (OZS) vs C and vs ZLS (LZS)), ** *p* ≤ 0.01 (ZFS (OZS) vs ZLS (LZS)), and ^*^*p* ≤ 0.05 (ZFS (OZS) vs. ZLS (LZS)).

**Table 1 ijms-20-04044-t001:** The untreated ZF4-SC and ZL4-SC from the pelvic muscles from 12 weeks old female ZF (Zucker fatty) and ZL (Zucker lean) rats have distinctive miR-GTS (global transcriptional signatures of microRNAs), which also differ from the miR-GTS in the untreated ED-MDSC (MDSC from young animals with mild early T2D/O) from the gastrocnemius of age-matched male ZF rats.

miR or Let	ZF4-SC-C	ZF4-SC-C/ ED-MDSC-C	ZL4-SC-C/ ED-MDSC-C	ZF4-SC-C/ ZL4-SC-C
ID	per 1000	Ratios		
**miR-143-3p**	**273.0**	**2.25**	**2.60**	**0.88**
**miR-99b-5p**	**204.0**	**1.49**	**0.88**	**1.69**
**miR-10a-5p**	**128.0**	**0.80**	**1.64**	**0.49**
**miR-21-5p**	**53.0**	**0.71**	**0.78**	**0.91**
**miR-100-5p**	**53.0**	**0.37**	**0.41**	**0.91**
**miR-99a-5p**	**25.0**	**0.52**	**0.21**	**2.50**
**miR-10b-5p**	**22.0**	**1.83**	**1.89**	**0.97**
**miR-125a-5p**	**18.0**	**2.65**	**0.67**	**3.95**
**miR-191a-5p**	**17.0**	**1.18**	**0.61**	**1.92**
**miR-26a-5p**	**17.0**	**0.67**	**0.32**	**2.12**
**let-7f-5p**	**17.0**	**0.53**	**0.32**	**1.67**
**miR-199a-5p**	**13.0**	**0.65**	**0.48**	**1.35**
**miR-221-5p**	**12.0**	**3.24**	**2.02**	**1.61**
**miR-222-3p**	**11.0**	**10.77**	**2.18**	**4.94**
**miR-27b-3p**	**9.6**	**0.80**	**0.21**	**3.87**
**miR-24-3p**	**9.3**	**1.03**	**0.57**	**1.81**
**miR-145-3p**	**8.2**	**1.91**	**1.50**	**1.27**
**miR-145-5p**	**8.0**	**2.53**	**4.07**	**0.62**
**miR-30a-5p**	**7.3**	**1.41**	**1.51**	**0.93**
**let-7i-5p**	**7.2**	**0.41**	**0.24**	**1.70**
**miR-148a-3p**	**6.2**	**1.17**	**2.06**	**0.57**
**miR-23a-3p**	**5.8**	**1.04**	**0.92**	**1.13**
**miR-30d-5p**	**5.7**	**1.10**	**0.63**	**1.74**
**miR-146b-5p**	**4.4**	**0.08**	**0.04**	**2.10**
**miR-151-3p**	**4.2**	**1.55**	**0.83**	**1.88**
**miR-199a-3p**	**4.0**	**0.36**	**0.31**	**1.17**
**miR-152-3p**	**3.8**	**0.38**	**0.39**	**0.97**
**miR-351-5p**	**3.2**	**2.11**	**0.55**	**3.83**
**let-7c-5p**	**3.0**	**0.57**	**0.24**	**2.41**
**miR-148b-3p**	**2.9**	**0.94**	**0.52**	**1.82**
**miR-22-3p**	**2.8**	**1.01**	**0.44**	**2.28**
**miR-221-3p**	**2.2**	**2.28**	**0.91**	**2.52**
**miR-27a-3p**	**2.1**	**1.08**	**0.76**	**1.42**
**rno-let-7g-5p**	**2.0**	**0.48**	**0.39**	**1.22**
**miR-181a-5p**	**1.8**	**1.64**	**1.56**	**1.04**
**miR-92a-3p**	**1.8**	**2.42**	**0.91**	**2.67**
**miR-342-3p**	**1.7**	**2.48**	**2.00**	**1.24**
**let-7b-5p**	**1.5**	**0.68**	**0.27**	**2.47**
**miR-23b-3p**	**1.5**	**0.61**	**0.24**	**2.51**
**miR-30e-5p**	**1.4**	**0.88**	**0.70**	**1.26**
**miR-7a-5p**	**1.4**	**2.21**	**4.94**	**0.45**
**miR-143-5p**	**1.2**	**2.71**	**2.18**	**1.24**
**miR-186-5p**	**1.1**	**1.07**	**0.70**	**1.54**
**miR-192-5p**	**1.1**	**1.45**	**0.62**	**2.35**
**miR-28-3p**	**1.1**	**1.27**	**1.01**	**1.26**
**miR-365-3p**	**1.0**	**1.44**	**1.17**	**1.23**

ZF4-SC-C, ZL4-SC-C, and ED-MDSC-C were the respective stem cells incubated in vitro as controls (C) without rat serum. The male ED-MDSC-C values per 1/1000 were previously reported [19] and not entered here. The top 47 miRs in the ZF4-SC-C expression values >10 per thousand of its total miRs were entered and sorted in descending order. Then, each miR value for ZF4-SC-C and ZL4-SC-C, respectively, was divided by the corresponding value in ED-MDSC-C as a reference to establish the respective ratios to the male control. The ZF4-SC-C to the ZL4-SC-C ratio was also established. Only the ratio values >1.95 or <0.54 were highlighted in blue or yellow, respectively, to compare changes between their respective miR-GTS.

**Table 2 ijms-20-04044-t002:** The short term in vitro exposure of ZF4-SC to a severely hyperlipidemic and mildly hyperglycemic serum (ZFS), induced considerable changes in the expression of specific miRs selected by their occurrence in the reference male MDSC and their relevance to myostatin, but not induced with the non-diabetic ZL serum (ZLS), or on the ZL4-SC under either type of serum.

ID for miR or Let	In Vivo	C Value per 10^6^ Reads	In Vitro Added 5% Serum (S)			
	MDSC		ZF4-SC		ZL4-SC	
	LD/ED Ratio		Ratios to C			
			ZFS/C	ZLS/C	ZFS/C	ZLS/C
**miR-21-5p**	**0.16**	**364070**	**0.52**	**1.17**	**0.69**	**1.13**
**miR-199a-5p**	**0.29**	**88010**	**0.41**	**0.86**	**0.86**	**0.97**
**rmiR-23a-3p**	**0.50**	**39134**	**1.11**	**1.09**	**1.14**	**0.86**
**miR-199a-3p**	**0.25**	**27501**	**0.41**	**0.86**	**0.86**	**0.97**
**miR-27a-3p**	**0.41**	**14228**	**0.99**	**1.08**	**1.02**	**1.16**
**miR-181a-5p**	**0.39**	**12538**	**0.54**	**0.94**	**1.66**	**0.89**
**miR-30e-5p**	**0.42**	**9387**	**0.44**	**1.14**	**0.63**	**0.94**
**miR-101a-3p**	**0.38**	**4728**	**1.06**	**0.97**	**1.05**	**0.98**
**miR-214-3p**	**0.22**	**4525**	**0.46**	**0.98**	**0.79**	**0.86**
**miR-29a-3p**	**0.25**	**2401**	**0.59**	**1.26**	**1.81**	**1.21**
**miR-101b-3p**	**0.09**	**804**	**0.49**	**1.13**	**0.86**	**1.17**
**miR-132-3p**	**0.02**	**340**	**0.68**	**0.99**	**1.46**	**0.90**

Values were selected based on miRs related to myostatin previously shown [18] to be up- or down-regulated by >2.0 in the late diabetes MDSC (LD-MDSC) from aged male ZF rats with the in vivo moderate-severe dyslipidemia, expressed as ratios vs. the control ED-MDSC from young male ZF rats with no dyslipidemia. This represented the impact of the long-term in vivo exposure of MDSC to the T2D/O milieu [19], as reference for the in vitro results, on the female stem cells. These miRs were entered in descending order of expression in the untreated female ZF4-SC-C. Both the ZF4-SC and the ZL4-SC were incubated in vitro with either hyperlipidemic ZFS obtained at 51 weeks of age from obese mildly hyperglycemic female rats, or from normolipidemic ZLS from normal weight normolipidemic age-matched rats, or no serum addition controls (C). The ratios between these miRs values in the treated stem cells and the ones in their respective controls are as tabulated. Only the ratios <0.54 were highlighted in yellow, with none >1.96. The green highlighted miR IDs indicate that miRs were affected similarly in the in vitro female ZF4-SC with ZFS and the reference in vivo male LD/ED-MDSC exposed to T2D/O.

**Table 3 ijms-20-04044-t003:** The in vitro miR transcription changes in Table 2 were also accompanied by changes in other individual miRs (assumed unrelated to myostatin), which were affected by the alterations induced in vitro by dyslipidemic serum.

ID for miR or Let	In Vivo	C Value per 10^7^Reads	In Vitro Added 5% Serum (S)			
	MDSC		ZF4-SC		ZL4-SC	
	LD/ED Ratio		Ratios to C			
			ZFS/C	ZLS/C	ZFS/C	ZLS/C
**miR-99b-5p**	**0.25**	**1393619**	**0.53**	**0.89**	**0.83**	**0.94**
**miR-10a-5p**	**0.28**	**871279**	**0.66**	**0.95**	**0.49**	**0.90**
**miR-100-5p**	**0.22**	**361258**	**0.63**	**0.78**	**0.64**	**0.84**
**miR-99a-5p**	**2.65**	**167053**	**0.37**	**0.95**	**0.83**	**0.85**
**miR-10b-5p**	**0.45**	**149991**	**0.33**	**0.92**	**0.46**	**0.84**
**miR-26a-5p**	**0.32**	**115483**	**0.99**	**1.12**	**1.89**	**1.04**
**let-7f-5p**	**0.32**	**115006**	**0.89**	**1.14**	**1.45**	**1.23**
**miR-221-5p**	**0.34**	**82561**	**0.59**	**1.26**	**0.96**	**1.19**
**let-7i-5p**	**0.12**	**49022**	**0.86**	**1.11**	**1.32**	**1.28**
**miR-148a-3p**	**0.25**	**42455**	**0.27**	**0.86**	**0.65**	**0.80**
**miR-152-3p**	**0.33**	**25783**	**0.42**	**0.97**	**0.88**	**1.01**
**miR-148b-3p**	**0.16**	**19779**	**0.51**	**1.06**	**0.98**	**1.26**
**let-7g-5p**	**0.22**	**13892**	**1.10**	**1.08**	**1.86**	**1.68**
**miR-92a-3p**	**0.17**	**12460**	**0.59**	**1.19**	**1.35**	**1.30**
**miR-342-3p**	**0.26**	**11421**	**0.22**	**0.99**	**0.48**	**0.84**
**miR-212-5p**	**0.08**	**4903**	**0.87**	**1.12**	**1.98**	**1.11**
**miR-10b-3p**	**0.10**	**4130**	**0.69**	**1.08**	**1.03**	**1.70**
**miR-25-3p**	**2.20**	**3984**	**0.67**	**0.94**	**1.10**	**1.34**
**miR-362-5p**	**0.20**	**1234**	**0.36**	**1.56**	**0.60**	**1.18**
**miR-31a-5p**	**0.29**	**1020**	**1.40**	**1.24**	**2.10**	**1.31**

See Table 2 for pertinent information, including highlighting in yellow. Highlighting in blue is for ratios >1.95. Most of these miRs have significance for stem cell biology.

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
