# Peer review of "Stem Cells from a Female Rat Model of Type 2 Diabetes/Obesity and Stress Urinary Incontinence Are Damaged by In Vitro Exposure to its Dyslipidemic Serum, Predicting Inadequate Repair Capacity In Vivo"

_ijms, 2019, doi:10.3390/ijms20164044_

Round 1
Reviewer 1 Report
For clarity, it is desirable that the distribution of animals (in vivo) and stem cells (in vitro) in groups.
Maybe the differences in the effects are associated with different tissue belonging to stem cells of females and males?
What explains the gender differences in the activity of SC gene mutations, or external mechanisms of regulation (hyperglycemia, dyslipidemia, sex hormones, inflammation features)?
Reviewer 2 Report
The study showed that muscle-derived stem cells are damaged by dyslipidemic serum. Thus, autograph stem cell therapy for FSUI may be ineffective in obese /diabetic patients. The study is well planned and conclusions are resonable, but the paper needs much improvements.
My comments and doubts:
Were the rats used in this study for MDSC isolation examined for the symptoms of FSUI?
There are many abbreviations in this paper, that is sometimes confusing.
All those abbreviations should be clearly explained. For example the ZF4-
SC, ZL4-SC, ZFS, ZLS abbreviations are well explained in methods section.
However, methods section in IJMS is at the end of the article. I suggest to move those explanation into the beginning of the results section.
The description of previous results in lines 76-79, page 2, is basically a repetition of the text from introduction (lines 57-62, page 2).
The meaning of markings in different colors in table 1 , table 2 and table
3 should be explained in table legends.
The description under table 1 (lines 102-110, page 3) is partly a
repetition of the text from lines 83-90 (page 2). Perhaps these two
fragments may be combined into one.
The dyslipidemia of ZFS serum used for MDCS treatment should be confirmed by at least basic tests such as triglyceride and cholesterol concentrations. The same for ZLS.
In lines 122 -125 authors wrote that table 2 presents 17 miR and 12 are
downregulated by ZFS treatment in ZF4-SC. However, in table 2 only 12 miR are presented and 7 are downregulated by ZFS treatment.
Line 170: I guess ZLC should be ZLS.
Figures 1D, 2D, 3D and 4D - what means "++++" on the graphs? The same for ** in fig 3D.
On Figure 3 D change "CTR LZS OZS" into "C ZLS ZFS" as in other figures.
ZLS induced apoptosis in ZL4-SC, but not in ZF4-SC. How would authors
explain this ?
Fig. 5A - again different abbreviations are used - ctr, lzs, ozs instead of
C ZLS ZFS. In the legend there is a mistake: panels B is not a micrograph,
whereas panel C is not a plot.
lines 251 - 254. Something is wrong with the formatting
The statistical significance is not presented on Figure 7
Line 300: Authors wrote: "The over-expression of myostatin by ZFS, but not
by ZLS, in the female ZF4-SC and ZL4-SC....." In my opinion ZLS also
significantly increases myostatin (Fig. 7 A-C) comparing to control, but,
as mentioned above significance is not presented on Figure 7.
